# Prosocial learning: Model-based or model-free?

**Parisa Navidi**[1]*, **Sepehr Saeedpour**[2], **Sara Ershadmanesh**[3,4], **Mostafa Miandari Hossein**[5], **Bahador Bahrami**[6]

1 Department of Cognitive Psychology, Institute for Cognitive Science Studies, Tehran, Iran, 2 Department of Electrical and Computer Engineering, University of Tehran, Tehran, Iran, 3 School of Cognitive Sciences, Institute for Research in Fundamental Sciences, Tehran, Iran, 4 Department of Computational Neuroscience, MPI for Biological Cybernetics, Tuebingen, Germany, 5 Department of Psychology, University of Toronto, Toronto, Ontario, Canada, 6 Crowd Cognition Group, Department of General Psychology and Education, Ludwig Maximilians University, Munich, Germany

* Parisanavd@gmail.com

**Data Availability Statement:** All data files are available from the OSF database (accession number(s) DOI 10.17605/OSF.IO/3K75V).

## Abstract

Prosocial learning involves the acquisition of knowledge and skills necessary for making decisions that benefit others. We asked if, in the context of value-based decision-making, there is any difference between learning strategies for oneself vs. for others. We implemented a 2-step reinforcement learning paradigm in which participants learned, in separate blocks, to make decisions for themselves or for a present other confederate who evaluated their performance. We replicated the canonical features of the model-based and model-free reinforcement learning in our results. The behaviour of the majority of participants was best explained by a mixture of the model-based and model-free control, while most participants relied more heavily on MB control, and this strategy enhanced their learning success. Regarding our key self-other hypothesis, we did not find any significant difference between the behavioural performances nor in the model-based parameters of learning when comparing self and other conditions.

## Introduction

There are times in everyone's life when the consequences of their choices are not limited to themselves. Depending on the situation, people voluntarily or involuntarily make decisions that influence others' advantage. Most of the time, decision-making and learning for self differ from what we do on behalf of others.

A wide range of previous studies have focused on the differences in decisions made for self and others. For example, proxy decision-makers (people who make decisions on behalf of someone else) solve a problem more wisely [1], have more creative ideas [2], and seek out more information [3]. Another line of research has shown that, in this process, individuals adjust their proxy choices based on the most prominent feature of in-hand issues, whereas for themselves, they consider various features evenly, giving nearly equal weight to all of them [4]. Also people change their mind more often in proxy decisions. More frequent switching can increase the probability of post-decisional distortion [5]. Making decisions for others, has been found to entail its own cognitive biases such as omission bias, pre-decisional distortion,

**Funding:** Bahador Bahrami is supported by the European Research Council (ERC), (https://argentumconsultants.eu/horizon-europe/?gclid=Cj0KCQiA4uCcBhDdARIsAH5jyUnW8TerqlCnpI6yo4A0CQjkVTrqVUJypT_8KowvXZDB6Jt81PcLJ7QaAtLjEALw_wcB) under the European Union's Horizon 2020 research and innovation programme (819040 - acronym: rid-O). Bahador Bahrami is also supported by the Templeton Religion Trust. The funders had no role in study design, data collection and analysis, decision to publish, or preparation of the manuscript.

**Competing interests:** The authors have declared that no competing interests exist.

confirmation bias, and lexicographic weighting [3, 6–9]. However, the consensus is that people often suffer fewer cognitive biases when they make decisions for other people [8]. These findings suggest that decisions for others might be more rational than those made for oneself.

Another factor that complicates prosocial decision making is that often, the other person for whom the decision is being made is present. One of the primary forms of intrapersonal effects discussed in social psychology is "social facilitation," which means that the presence of others can enhance or impair an individual's performance [10]. Investigating a set of studies, Zajonc has shown that the presence of others, either as observers or colleagues, could enhance participants' performance but would disrupt the process of learning new and complex issues [11]. More recently, Kumano and colleagues demonstrated that, in a value-based decision-making context–In value-based decision-making we choose one option out of many that have different intrinsic values for us. In another world our idiosyncratic values determine which option to choose [12], when participants played on behalf of their partner, in the presence of the partner they took less reasonable risks, and were more affected by anticipated regret [13].

Being watched by others will lead to cardiovascular changes [14], and stimulate social anxiety–a severe fear of negative evaluation by others–often causing people to avoid social interactions [15], which can explain the changes in behaviour and disruption of the learning process observed by the other studies listed above. Schwabe and Wolf showed that being imposed by stress, here, socially evaluated cold pressor (SECP), before and after learning, leads people to rely more on habitual (model-free) learning [16]. Based on Gilbert's social rank theory, fear of negative evaluation of others can increase the level of subordinating behaviour among those suffering from this type of social anxiety [17]. As a result, under the pressure of social evaluation, people may apply more implicit and automatic responses representing a direct effect of social anxiety in the learning mechanism [18]. As regards risk-taking, although there is a debate on this issue, Polman and Wu have found that people make riskier decisions for others than themselves [19] which can increase the stress in proxy decision-making. Together, the available evidence suggests that the presence of the beneficiary may increase the cognitive burden of making choices for others.

Whereas self-other decision-making have been extensively studied, far less is known about self-other and prosocial learning. There are a number of recent studies [20–23] that have focused on the question of how we learn the preferences of others, in order to make decisions for them. It is important to clarify here that by prosocial learning we refer to learning *on behalf of* someone else to benefit them, and not learning *about* someone else. In other words, our question here can be described as how do we learn when our decisions benefit someone else.

To answer this question, here we focus on an experimental paradigm from the family of two-system theories of learning [24, 25]. The overarching theme of these models of learning is that learning is governed by two systems. One is fast, automatic, rigid, and model-free, and the other is slow, deliberative, flexible, and model-based, [26–29]. These two systems are always in competition to control our actions [30]. This is not to say that one of these systems is constantly dominant or brings us more benefits all the time. In fact, the benefit of having two different systems for control is to achieve behavioural goals by exploiting the system that best matches the moment-to-moment requirements of the environmental conditions [31]. A computational approach [30] has implemented this two-system model within the reinforcement learning paradigm [32].

At the simplest level, an agent could learn from repeating the actions that previously led to the greatest reward. This would constitute model-free (MF) learning [33]. A more sophisticated model-based (MB) strategy would be to estimate the future outcome of each action according to a learned transition structure of the environment (world model) and, as a result, chooses the action that promises highest reward within that world model [28, 32, 34]. This flexible model-based strategy will be updated by environmental changes [35]. When the action

supported by the MF and MB strategies are in conflict, the system whose prediction for outcome has a higher precision dominates the behaviour.

Searching through all the possible ways to learn the environment is costly and, in some cases, impossible [27, 31]. This means the MB strategy is flexible but slow and cognitively demanding. On the other hand, if the world is stable for long periods of time, the MF strategy is computationally optimised since prediction error would be progressively decreased. However, more trial and error would be needed in this strategy as the estimated values are not flexible [36]. The MF strategy is cognitively cheap but slow in responding to environmental changes. In complicated situations, where more planning will be needed the model-based system is preferred [31]. For example, at the beginning of the learning process, individuals rely more on a model-based strategy to identify the environment. As time goes by, as they get more and more experience in the same environment, behaviour becomes more habitual or automatic [37]. Thus, learning behaviour is often best described by a mixture of MB and MF [38].

MB decision-making costs mental effort [39] but is likely to result in a more desirable outcome in the future [38]. On the other hand, when under pressure to perform, people tend to decrease MB control and employ the low-demand MF alternative [40]. As children get older they learn the transition structure of the environment better and demonstrate more MB behaviour [41]. Similarly, under high cognitive load [42] and acute stress [43–45] people shift to MF control. Bringing these ideas together, here we ask whether the balance between MF and MB strategies changes when we are learning to make value based decisions for ourselves vs. on behalf of others i.e., prosocial learning.

To answer this question, we implemented a canonical example of the reinforcement learning paradigm configured to optimally assess MB and MF learning [38]. Participants undertook the task once for themselves and once for another person. To create a strong social context, under both conditions, the other person was also present in the experimental session and evaluated the participant's performance. If decisions for others were made more rationally with higher cognitive effort, we predict that learning for others should tip the balance in favour of the MB strategy. On the other hand, if deciding for others increased cognitive load and induced social stress, then one we expect learning to favour MF strategies.

## Methods

### Participants

Seventy-six healthy individuals (34 female) with different levels of education participated in this study between December 2020 and September 2021. They were recruited through local advertising in the campus area. Participants were randomly assigned to a pair (19 pairs: female-male, 8 pairs: female-female, and 11 pairs: male-male) and scheduled to attend the same test session. Members of each pair did not know each other. To determine sample size, we decided based on previous studies with a total sample of 38 players.

### Inclusion criteria

We recruited individuals between the ages of 18 and 35 who had no history of neurological or psychiatric disorders and no prior participation in studies related to value-based decision-making.

### Exclusion criteria

We excluded participants who missed more than 20% of the trials, as well as those who consistently selected the same option throughout the experiment or indicated in the post-experiment

debriefing that they did not understand the task structure. Following these exclusion criteria, we removed two participants from the initial sample, resulting in a final group of 36 players (18 females; age: 24.39 ± 2.61, mean ± STD) for data analysis.

All participants filled in an informed written consent form, approved by the Iranian Institute for Cognitive Science studies Ethics Committee (approval ID IR.UT.IRICSS. REC.1399.005).

## Apparatus and setup

All experiments were conducted in a 10 m$^2$ conference room at the University of Tehran, College of Engineering. Participants sat at a table next to each other about 1 metre apart. There was a 17-inch laptop in front of one participant (i.e., the player) to which the other participant (i.e., the observer) had a clear view. The players completed a two-step task designed by Kool, Cushman, & Gershman, 2016, [38]. This paradigm allowed us to distinguish the MB and MF behaviour among subjects. Based on our pilot results, all task parameters matched the original design except the response time, which we decreased from 2000ms to 1000ms. We designed the task in MATLAB using Psychtoolbox (http://psychtoolbox.org/).

## Procedure

Upon arrival, the participants were greeted by the experimenter who gave them a brief explanation about the procedure and assigned the participants' roles in the experiment by an ostentatious coin toss. One participant was assigned to the role of the player and the other to that of the observer.

The experimenter then described the task instruction for both individuals. The experiment consisted of two rounds. In one round, the player did the task for themselves, meaning whatever bonus the player managed to collect would be paid to them. While the player was engaged in the task, the observer's task was to follow the player's performance carefully without talking to them. The observer filled in a paper-based questionnaire to evaluate the player's performance. These evaluations were not included in the data analysis. Instead, they served to reinforce the evaluative role of the audience and raise the stakes for the player. The experimenter was present in the room throughout the experiment.

The experimenter then described the gamified structure of a learning task for both of the player and the observer with this story: "You will play a game of intergalactic trade with aliens that live on other planets. In each trial, you load your goods on a spaceship and send it to some target planet. In each trial, there are two spaceships you can choose from. The chosen spaceship will then travel to one of two planets. Your profit or loss will depend on the value of your goods on the destination planet. At any given point, your goods are more valuable on one of the two planets. Your job is to figure out which spaceship would take your goods to the more profitable planet. The more profit you make, the higher your bonus. Note that this profitability changes across time and you will have to switch from one planet to another every once in a while."

The experiment began after the experimenter ensured that the participant had understood the task instruction adequately, starting with 25 practice trials to become familiar with the experiment. After practice, in half of the groups, the player did 126 trials in 3 blocks for themselves first and then another 126 trials for the observer. Other half of the groups the player started by playing on behalf of the observer. At the start of each block, the experimenter reminded the participant whether they were playing for themselves or the observer. Furthermore, participants were informed that the figures and rules were identical in both the self and other conditions, except for the colour of the destination planets. In half of the groups, yellow

and green planets were used for the self condition and red and blue planets for the other condition, while in the other half, red and blue planets were used for the self condition and yellow and green planets for the other condition. At the end of blocks, participants took a short break. After each condition, they took a 10-minute break. There was a fixed compensation for each participant. In addition, the player received a variable compensation based on the sum of points that the player earned under the "self" condition. The observer receiving a variable compensation determined the points participant's bonus was negative, it was set zero, and the participant received the fixed compensation.

## Design

The task design closely followed an earlier work (Kool et al). Each trial consisted of two stages. In the first stage, the participant was given two spaceships to choose from. In the second stage, the chosen spaceship travelled to its destination and the corresponding profit or loss was revealed. All in all, there were two pairs of spaceships (S1-S2 and S3-S4) and two planets (yellow and green or red and purple, depending on the conditions) in the experiment. Note that S1 was always paired with S2 and S3 always with S4. Spaceships S1 and S3 both went to the yellow planet. S2 and S4 went to the green planet. The spatial position (left or right) of the spaceships on the screen were randomly assigned across trials. By pressing the F or J buttons on the keyboard the participant could select the left or right option, respectively. Response time window was 1000ms. Otherwise, the trial was missed. After selecting their spaceship, the participant pressed the spacebar to proceed to the second stage and get the profit or loss which was a number between -4 to +5. Thus, each trial had a definite correct choice: the spaceship that returned the higher payoff (Fig 1).

In trial t, when a subject selects one of the spaceships from a given pair, the outcome from the associated planet may not necessarily affect their choice in the following trial (t+1) when presented with a different pair of spaceships. This type of behaviour is referred to as model-free (MF) behaviour. On the other hand, it is also possible for subjects to exhibit a behaviour called model-based (MB), where subjects use their knowledge of the task structure to make predictions about the outcome of their choices, and generalise the outcome from the chosen

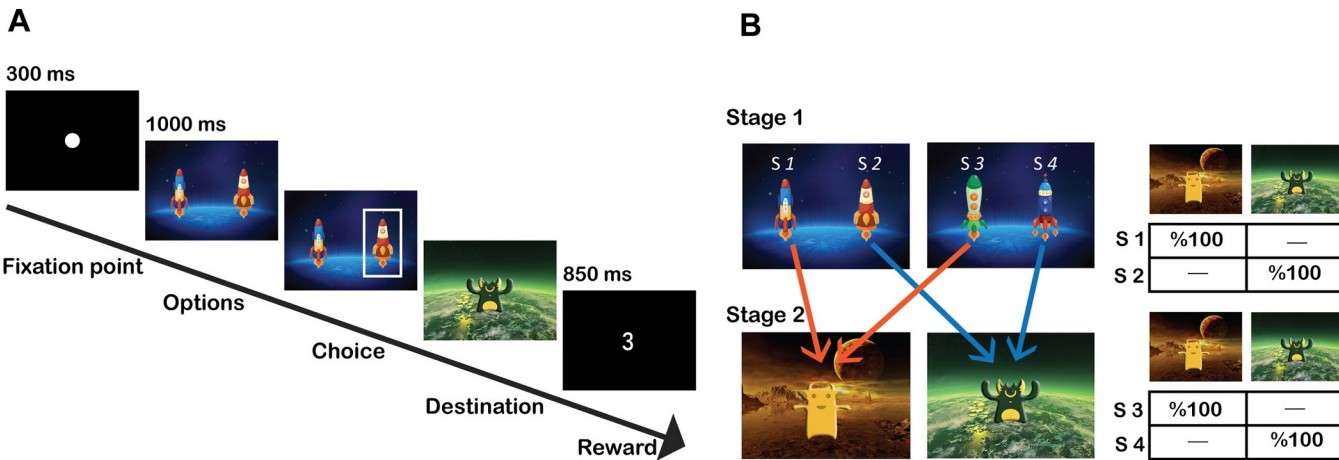

**Fig 1. Experiment design.** (A) The experiment process: In the first stage, participants choose one of the spaceships, and the selected spaceship is deterministically transferred to its destination planet. In the second stage, by pressing the space key, the payoff of this choice was shown, this outcome changed slowly based on a random Gaussian walk. (B) Law of transition: There are two pairs of spaceships. Each spaceship in a pair has a fixed destination and an equivalent spaceship in the other pair.

spaceship to the spaceship in the new, different pair that leads to the same planet. In other words, a MB agent's performance often does not change when they are faced with different stimuli. We used the word "stay" to refer to a kind of action where participants repeated their previous choice, either from the identical stimulus or from a different stimulus by selecting the equivalent spaceship (a spaceship in a different pair but with an identical planet destination).

For each participant, a new set of payoffs was generated. We used a random walk with a Gaussian distribution ($\mu = 0$, $\sigma = 0.2$) in a range of 0 to 1. A set of probabilistic numbers determined the outcome of planets. Based on the mean of previous outcomes ($\mu$), a random number was added or subtracted from the generated reward. Sigma shows the variance of changes in each trial. To be more sensible, following Kool et al (2016), the outcomes were displayed as integers between -4 to +5. The spaceships, rules of transition, and reward generation systems were the same in all conditions of the task (playing for self, for the other, and in the practice phase), but the colour of the planets was different.

It should be noted that, before implementing the main experiment, we ran three pilots with a total of 22 participants to ensure that we could replicate the original findings (Kools et al, 2016) that would allow us to differentiate between model-based and model-free behaviour. The results of the pilots guided us in two ways: firstly, we changed the number of trials slightly from the original analysis to fit our design. Secondly, we observed in our pilots that behaviour was dominated by the MB system. To achieve a more balanced combination of MB and MF systems, we reduced the response time from 2000ms to 1000ms to add some urgency [40]. We ran the main experiment when the pilots' results confirmed that this paradigm worked well.

## Computational modelling

A reinforcement learning model was fitted to each participant's data to estimate the weighting parameter *w* (the relative contribution between model-based and model-free control). Each participant's weighting parameter in the new paradigm was calculated according to the original reinforcement learning model, but the transition structure was estimated differently. A pure model-free agent is not affected by the reward obtained from one pair of spaceships when choosing between the different pairs on the subsequent trial. Therefore, as in prior studies, the SARSA($\lambda$) temporal difference learning algorithm defines MF learning. As mentioned, since the transition structure is deterministic in the novel paradigm, there were some differences in the learning transition structure for a MB agent compared to the original paradigm. The second stage outcome influences the following choice of a pure MB agent, regardless of whether the first stage begins with the same pair of spaceships or the other pair. Model-based learning was calculated within Bellman's equation using the softmax decision rule. It defines the transition structure P in which the probability of action 'a' on trial 't' in the state 'i' is defined as below:

$$P\big(a_{i,t} = a | s_{i,t}\big) = \frac{\exp(\beta[Q_{net}(S_{i,t},\ a) + \pi.rep(a) + \rho.resp(a)])}{\sum_{a'}\exp(\beta[Q_{net}(S_{i,t},\ a') + \pi.rep(a') + \rho.resp(a')])}$$

Here the inverse temperature $\beta$ indicates the randomness of choice. The "stickiness" parameter $\pi$ measures the extent to which participants persisted ($\pi > 0$) or switched ($\pi < 0$) their choice at the first stage. The variable rep($\alpha$) is set to 1 if the chosen option at the first stage ($\alpha$) is the same as the previous trial; otherwise, it is set to zero. "Response stickiness" parameter $\rho$ measures the extent to which participants repeated ($\rho > 0$) or changed ($\rho < 0$) the key that they had pressed at the first stage compared to the previous first-stage trial. The variable resp($\alpha$) is set to 1 if the selected key at the first stage ($\alpha$) is the same as the selected key on the previous trial; otherwise, it is set to zero. Following the reference work (Kool et al), maximum 'a

posteriori' estimation with empirical priors was used. We also set our model's free parameter based on our reference: inverse temperature, $\beta \sim$ Gamma (4.82, 0.88), and stickiness parameters, $\pi, \rho \sim$ N (0.15, 1.42), and flat priors for all other parameters.

## Results

To see if our experiment worked properly, we compared the predicted behaviour from Kool's task, as a reference, with our results. As Kool and colleagues had shown in the novel paradigm, participants tended to rely more on model-based than model-free strategy, we saw a similar pattern. While some participants showed a combination of these two strategies, a great number of them had a pure MB behaviour. Regardless of the condition (the player did the task for themselves or did on behalf of the observer), the results showed the mean of stay probability was almost as much as our reference ones (~ 0.8). Using a 2*2*2 ANOVA test, we found a significant main effect for outcome valence [$F(1,35) = 246.45$, $p<0.001$] and interaction between outcome valence and stimulus [$F(1,35) = 6.80$, $p<0.013$]. In other words, for both the same and different stimuli, the stay probability significantly increased when the previous outcome was rewarding compared to when it was punishing. All of these results confirmed that the basis of our task worked correctly.

Although stay probability in other conditions for both outcome valence (mean reward = 0.81 and mean punishment = 0.44) was slightly different than that in self rounds (mean reward = 0.85 and mean punishment = 0.45); this difference was insignificant, ($t = -1.31$, $df = 35$, $p = 0.17$, CI = [-0.27, 0.06]) for reward outcome and ($t = -1.31$, $df = 35$, $p = 0.17$, CI = [-0.27, 0.06]) for punishment outcome, (Fig 2A). We also evaluated the difference between some behavioural parameters in self and other conditions. Based on the two-tailed paired t-test results, there was no significant difference in these measures between self and other conditions. For example, we did not see any significant difference between accuracy (the number of correct choices), ($t = -0.65$, $df = 35$, $p = 0.52$, CI = [-0.04, 0.02]), response time, ($t = -0.58$, $df = 35$, $p = 0.57$, CI = [-0.02, 0.01]), and relative performance (sum of points a subject gained, divided by maximum attainable reward), ($t = -1.31$, $df = 35$, $p = 0.17$, CI = [-0.27, 0.06]).

Following Kool et al. study (2016), a multilevel logistic regression was fitted to the data. All coefficients were modelled as random effects to estimate individual differences in choice

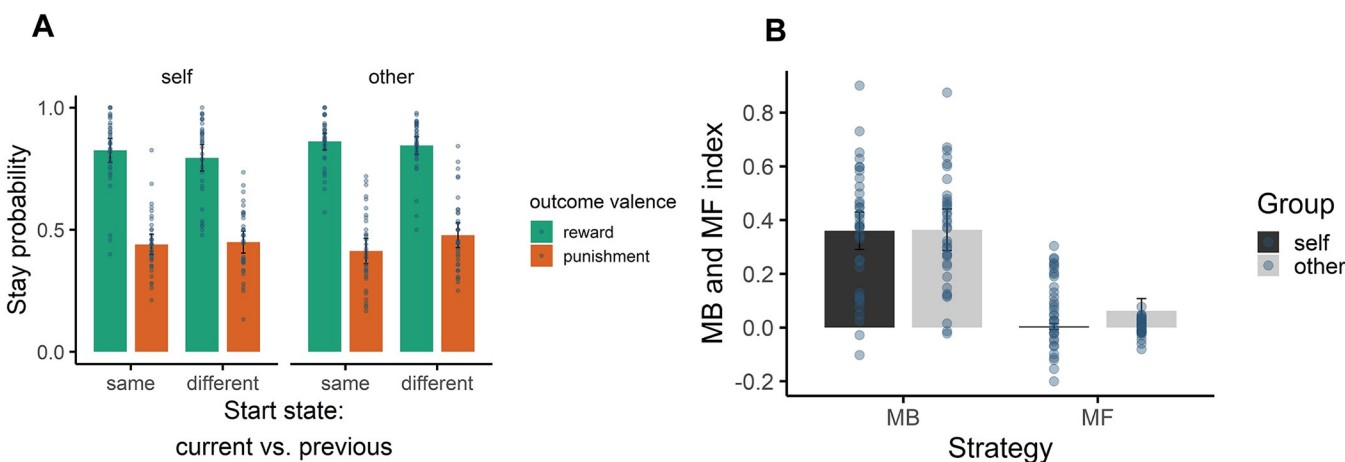

**Fig 2. Behavioural performance and model-based and model-free indices in self and other rounds.** (A) The horizontal axis shows whether the current state started is the same or different compared to the previous start state; the vertical axis shows the probability of repeating the choice, which leads to the same outcome as the previous stage. (B) The model-based and model-free index in self and other conditions based on the multilevel logistic regression analysis. The model-based control was the dominant behaviour among participants, while there was not a significant difference between these strategies in self and other conditions.

**Table 1. Regression coefficients from multilevel logistic regression analysis in self conditions.**

|  | Estimate | Std. Error | z value | Pr(>|z|) |
|---|---|---|---|---|
| (Intercept) | 0.497524 | 0.089217 | 5.577 | 2.45e-08 *** |
| same | 0.001122 | 0.049351 | 0.023 | 0.982 |
| diffrence | 0.082914 | 0.019900 | 4.167 | 3.09e-05 *** |
| reward | 0.372286 | 0.044298 | 8.404 | < 2e-16 *** |
| same:difference | 0.003216 | 0.019600 | 0.164 | 0.870 |
| same:reward | 0.003794 | 0.022754 | 0.167 | 0.868 |
| difference:reward | -0.004444 | 0.005917 | -0.751 | 0.453 |
| same: difference:reward | 0.007198 | 0.006117 | 1.177 | 0.239 |

behaviour for both self and other conditions (Table 1). Then the results for the two conditions were compared to each other. In this model, the dependent variable was whether an individual repeated or changed the current choice compared to their previous choice. Predictors for each trial were the amount of points gained on the previous trial (reward$_{i-1}$) and whether the current trial started with an identical (same$_i$ = 1) or different (different$_i$ = 0) pair of spaceships compared to the previous trial. A third predictor was the difference between the outcome of chosen and unchosen options in the previous trial (difference$_{i-1}$). Although subjects did not observe the outcome of the unchosen option, Kool et al added this predictor as a parallel term to the weiting parameter and due to the autocorrelation of outcomes for each planet across trials, the current unchosen outcome can be considered a suitable estimate of the last reward observed by the subjects. Regardless of the stimulus (same or different), gaining a reward is the most important measure for a MB agent. In this model, the main effect of the reward in the previous trial (reward$_{i-1}$) would indicate MB choice behaviour. On the other hand, a different stimulus will not affect a MF agent. Therefore, the participant would not necessarily choose the correct option when facing a different pair of spaceships. As a result MF control is represented by the interaction of the previous reward and the regressor for same transition (reward$_{i-1}$ * same$_i$).

As Fig 2B shows, the MB control was the dominant strategy in both self (mean = 0.360) and other conditions (mean = 0.364), but there was no significant difference between subjects' MB index in playing on self and other rounds (t = -0.09, df = 35, p = 0.93, CI = [-0.09, 0.08]). It was true that there was a slight difference in MF control between self and other conditions (t = -2.61, df = 35, p = 0.01, CI = [-0.10, -0.01]), but we ignored this difference since this strategy was less dominant among participants.

In addition we looked at the relationship between the MB and MF indexes and participants' performance. Following Kool's paper as a guideline [38], we expected participants' performance to improve when they applied a more MB approach. Pearson correlation results showed that both MB and MF strategies affected one's performance in an opposite direction. The more MB control a person implemented, the higher performance she or he would reach. Following MB strategy significantly correlated with choosing more correct answers, (Pearson correlation = 0.63, p = 001, 95%CI = [0.46, 0.75]), (Fig 3A), on the other hand, people with high score in MF control significantly showed lower performance, (Pearson correlation = -0.29, p = 013, 95%CI = [-0.49, -0.06]), (Fig 3B). This was another sanity check which approved that we have correctly conducted our experiment.

One of the interesting points was that the Factorial repeated measure ANOVA analysis showed that MB control more affected the performance in self conditions than in other conditions (F (3, 68) = 14.49, p = 0.0001), but since this was not related to our question, it can be addressed in the future studies (Fig 3C).

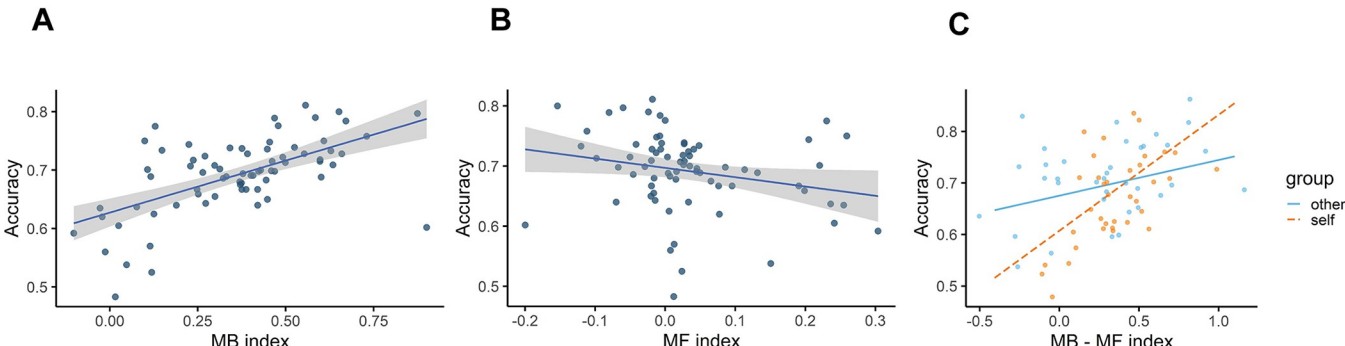

**Fig 3. Correlation between MF / MB and performance.** (A) The effect of model-based and (B) model-free behaviour on participants' accuracy. The X-axis represents the model-based/free indexes, and the Y-axis represents the mean of accuracy for every subject. (C) The effect of model-based control Factorial repeated measure ANOVA analysis showed that the positive effect of model-based behaviour on performance in self conditions is higher than that on performance in other condition.

## BFNE-II questionnaire

Our participants played the game while their performance was being observed and evaluated by their teammate who had the observer role; therefore, we asked all players to fill the Bfne-II questionnaire (a brief fear of negative evaluation questionnaire [46]) on an online platform. The Pearson correlation analysis showed that there was no significant correlation between MB control ((r = 0.10, p = 0.581, 95% CI = [-0.25, 0.43]); (r = 0.0, p = 0.992, 95% CI = [-0.34, 0.34]) in self and other conditions, respectively) or w parameter ((r = 0.03, p = 0.867, 95% CI = [-0.32, 0.37]); (r = -0.05, p = 0.775, 95% CI = [-0.39, 0.30]) in self and other conditions, respectively) and social anxiety score in this study.

## Computational modelling

Pearson correlation analysis showed that the corrected reward rate (the difference between average reward distribution of each participants and their reward rate) positively correlated with the weighting parameter, (Pearson correlation = 0.66, p = 001, 95%CI = [0.43, 0.81]; = 0.38, p = 0.02, 95%CI = [0.06, 0.63]) in self and other conditions, respectively, (Fig 4A). There was a positive correlation between the *w* parameters estimated by model fitting and the MB parameters resulting from regression analysis on behavioural data (Pearson correlation = 0.63, p < .001, 95%CI = [0.46, 0.75]). Linear regression results confirmed this observation as well (F (1, 34) = 27.04, p = 0.0001). However, the effect of MB behaviour on performance or reward rate did not differ between self and other conditions. We also did not see any significant difference between weighting parameters in self conditions and other conditions (t = -1.39, df = 35, p = 0.17, CI = [-0.18, 0.03]), (Fig 4B).

## Discussion

We took a classical reinforcement learning paradigm in value-based decision making to a social context and asked whether model-based and model-free strategies control learning differently when people learn on behalf of themselves vs. for others i.e., prosocial learning [47]. Our paradigm was designed to tease out the contribution of the model-based and model-free controllers [38]. We replicated the key hallmarks of the previous findings. Critically, we observed that a larger proportion of our participants employed the MB strategy successfully. Moreover, consistent with previous findings, we observed that the greater the degree of MB control, the higher the participants' accuracy in learning. These findings lend

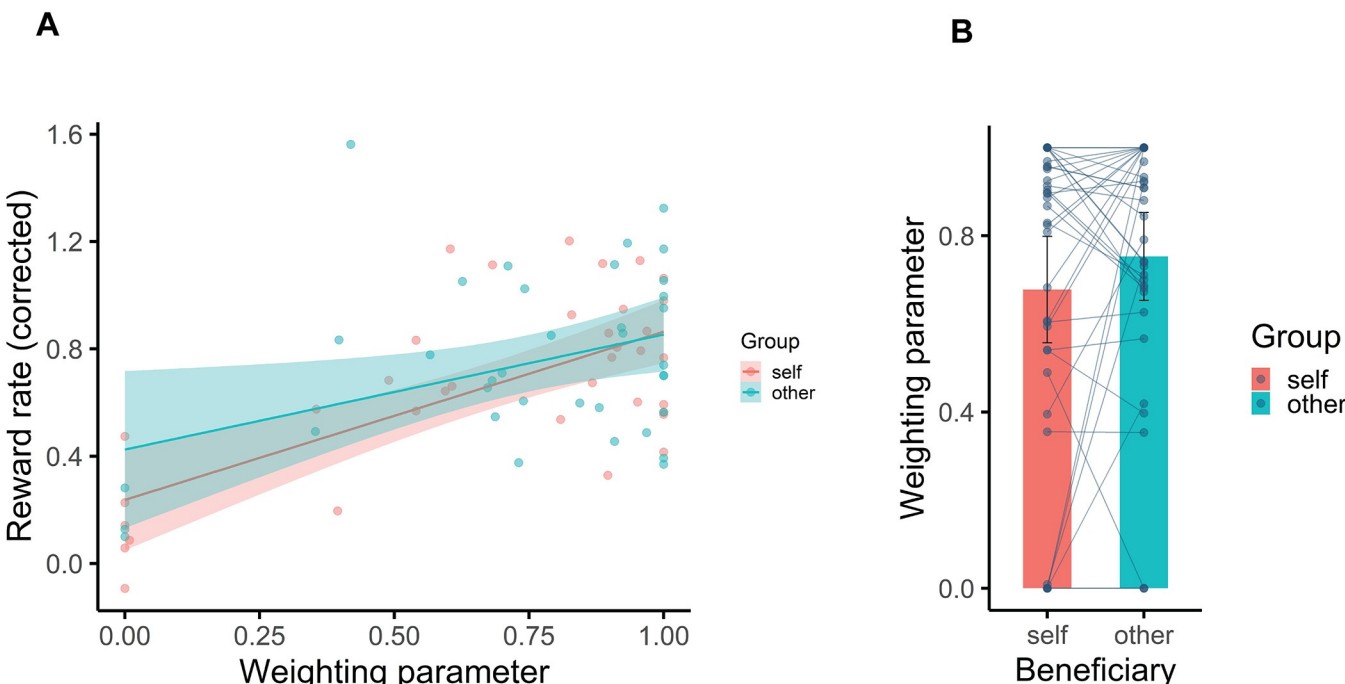

**Fig 4. The results of model fitting.** (A) The effect of weighting parameters on participants' accuracy. The X-axis represents the weighting parameter, and the Y-axis represents the corrected reward rate. The more model-based behaviour participants had, the more reward they gained on either condition. (B) Weighting parameters in self and other conditions. Although the *w* parameter was higher when participants played for the observer compared to doing it for themselves, this difference was not significant.

support to the validity of our paradigm and permitted us to proceed to testing our main hypothesis.

Overall, we did not see any difference between participants' behavioural data (such as accuracy, response time, and performance) and their learning strategy for themselves vs. others. Participants made value-based decisions, in some blocks for themselves and in other blocks for their teammate who was present in the testing room and sitting next to them watching over them and conspicuously evaluating the participant's performance. This manipulation, however, did not affect the participants' learning strategy remarkably. We report this negative result to avoid the "file drawer" bias–the tendency to publish the studies that show significant results [48].

In one exploratory analysis, we found the positive effect of MB control on performance accuracy in learning for self and others. However, in self conditions this effect was stronger than that in other conditions (Fig 3C). This observation is in line with the lack of a difference in average MB index and average performance. However, it is not entirely clear to us why this interindividual relationship between learning strategy and performance accuracy should break down while deciding for others. Future computational studies that examine the role of context (such as deciding for self or other) on control mechanisms in learning could address this problem.

Our results did not show any significant difference between learning strategies when learning on behalf of self vs. others. There could be a number of reasons for these negative results. We had made a design choice to keep the partner present in the testing room hoping to exaggerate the impact of our social manipulation. We had predicted that decision making for others (vs. self) is more likely to follow the MB strategy and the presence of the partner will

strengthen this tendency by heightening the participant's level arousal and attention. If, however, the impact of deciding for others were to decrease the MB learning and the presence of the partner to enhance performance, then it could be imagined that the presence of the partner might counteract and cancel this effect of making decisions for others. In addition, participants were informed about the beneficiary (i.e., self or other) they were playing for at the beginning of each block, and were not reminded of this during the blocks. Another previous study of prosocial learning [47] used a trial-by-trial repetitive reminder. One may argue that if the reminder is only shown at the beginning of the block, this would implicate greater cognitive load imposed on the participant, requiring them to be continuously reminding themselves who the money goes to. This could have piled up on the social observation and result in added pressure. Thus, under this cognitive load, people may have quickly forgotten about beneficiary and regressed to performing the task under the same condition irrespective of who the beneficiary was. The block design of the repetition of the instructions could have imposed an additional cognitive load on our participants, and should be taken into consideration in future studies.

A key factor in our hypothesis for a difference between deciding for self and others was that previous studies had indicated that when the task at hand requires high cognitive load, then participants do better when deciding for others. To ensure that our task's required cognitive load was adequately high, we performed several pilots and implemented the task with a limited time-window for response to increase the pressure on the participant for performing well. Our behavioural results (Figs 2 and 3) confirmed that our participants were adequately far from perfect to rule out ceiling performance. The nature of our negative findings, however, leave the possibility open that manipulation of effort may not have been adequately effective.

An important distinction in terminology will help us see the relationship between the current work and the previous woks in the literature. When one's actions benefit someone else, that has been called prosocial learning [47]. Another similar term that has been used in contexts of observational learning is vicarious learning which refers to learning from observing other's actions and outcomes, without performing the actions oneself [49]. The current study focuses on the former, where an individual makes choices and observes their outcomes, but sometimes the monetary benefit of the outcomes will go to another person instead of oneself. In other words, a relevant distinction for the processes of social reinforcement learning is whether the individual makes the choice/action themselves or only observes the choice/action made by another. This is important because previous studies have shown that these differences engages different computational [50] and neural substrates [51]. Here, the key difference between the conditions is whether the participant believes that optimising their choices to make the most points will eventually result in more money for themselves or for another person i.e., the physically present confederate. Importantly here, in both conditions the participants get the same information from making choices and observing their outcomes. Indeed, the making of the choice [52] and the information gained from the action and the feedback are intrinsically rewarding [53]. The monetary gains in experiments like ours is a bonus that (we experimenters hope and believe) motivates the participant to do even better. As such, in our study, participants did not engage in vicarious learning. However, one could argue that, circumstantially, they did process vicarious information through observing the benefit that went to their confederate, albeit caused by the participant's own actions and choices. To summarize, what makes our study different from those studying vicarious reward is that in those studies, individuals learned from observing others' actions and outcomes without taking any actions themselves [49, 54]. However, in our study, participants' learning occurred by taking some action to benefit another, which is called prosocial learning [47]. Previous studies have demonstrated a causal relationship between social behaviour and factors such as perceived similarity

[54], vicarious anxiety [55], and empathy [47]. In our study, the player and observer were seated next to each other, which could have manipulated these factors and potentially explains why we did not observe any differences between the self and other conditions.

Finally, our participants did not know each other and members of each pair were recruited independently from one another. A previous report that demonstrated different levels of antic-ipated regret and risk taking when making decisions for others [13] recruited pairs who were familiar with one another. We hope that our study and the report of the negative findings helps future researchers who are interested in similar questions regarding prosocial learning and decision making.

## Author Contributions

**Conceptualization:** Parisa Navidi, Sara Ershadmanesh, Mostafa Miandari Hossein, Bahador Bahrami.

**Data curation:** Parisa Navidi.

**Formal analysis:** Parisa Navidi, Sepehr Saeedpour, Sara Ershadmanesh.

**Funding acquisition:** Bahador Bahrami.

**Investigation:** Parisa Navidi.

**Methodology:** Parisa Navidi, Bahador Bahrami.

**Project administration:** Parisa Navidi, Bahador Bahrami.

**Resources:** Parisa Navidi.

**Software:** Parisa Navidi, Sara Ershadmanesh, Mostafa Miandari Hossein.

**Supervision:** Bahador Bahrami.

**Validation:** Parisa Navidi.

**Visualization:** Parisa Navidi.

**Writing – original draft:** Parisa Navidi.

**Writing – review & editing:** Parisa Navidi, Bahador Bahrami.

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
