## [Decision Letter · Decision Letter 0]

5 Apr 2023

PONE-D-22-34158

Vicarious learning: model-based or model-free?

PLOS ONE

Dear Dr. navidi,

Thank you for submitting your manuscript to PLOS ONE. After careful consideration, we feel that it has merit but does not fully meet PLOS ONE’s publication criteria as it currently stands. Therefore, we invite you to submit a revised version of the manuscript that addresses the points raised during the review process.

We look forward to receiving your revised manuscript.

Kind regards,

Francesca Benuzzi, Ph.D.

Academic Editor

PLOS ONE

“Bahador Bahrami is supported by the European Research Council (ERC), (https://argentumconsultants.eu/horizon-europe/?gclid=Cj0KCQiA4uCcBhDdARIsAH5jyUnW8TerqlCnpI6yo4A0CQjkVTrqVUJypT_8KowvXZDB6Jt81PcLJ7QaAtLjEALw_wcB) under the European Union’s Horizon 2020 research and innovation programme (819040 - acronym: rid-O). Bahador Bahrami is also supported by the Templeton Religion Trust.”

“BB is supported by the European Research Council (ERC) under the European Union’s Horizon 2020 research and innovation programme (819040 - acronym: rid-O). BB is also supported by the Templeton Religion Trust.”

“Bahador Bahrami is supported by the European Research Council (ERC), (https://argentumconsultants.eu/horizon-europe/?gclid=Cj0KCQiA4uCcBhDdARIsAH5jyUnW8TerqlCnpI6yo4A0CQjkVTrqVUJypT_8KowvXZDB6Jt81PcLJ7QaAtLjEALw_wcB) under the European Union’s Horizon 2020 research and innovation programme (819040 - acronym: rid-O). Bahador Bahrami is also supported by the Templeton Religion Trust.”

Additional Editor Comments:

The paper presents an interesting and complex study but needs some revisions to be ready for publication.

Reviewer 1 highlighted some minor issue in the text.

Reviewer 2 is positive but asked for major revision.

I agree with Reviewer 2 that data partially support discussion; some theoretical aspects need clarification and results section needs clarification since is quite difficult to follow.

Reviewers' comments:

Reviewer's Responses to Questions

**Comments to the Author**

1. Is the manuscript technically sound, and do the data support the conclusions?

Reviewer #1: Yes

Reviewer #2: Partly

2. Has the statistical analysis been performed appropriately and rigorously? 

Reviewer #1: Yes

Reviewer #2: I Don't Know

3. Have the authors made all data underlying the findings in their manuscript fully available?

Reviewer #1: Yes

Reviewer #2: No

4. Is the manuscript presented in an intelligible fashion and written in standard English?

Reviewer #1: Yes

Reviewer #2: Yes

5. Review Comments to the Author

Reviewer #1: Overall, the manuscript is well-written and clear. Below some feedback on specific sections:

Line 62: it would be helpful if the authors define what a value-based decision is, as this is not clear from the context.

Line 151: mention inclusion and exclusion criteria specifically.

Overall feedback on Discussion: the discussion is well-written as far as analysing the findings of this study, however what seems to be missing is placing these findings in the context of previous literature. There are also no references to previous work. It would strengthen the paper to discuss the findings in light of similar studies on vicarious learning.

Line 410: elaborate on the ‘file drawer’ bias and add a reference.

Minor comments on grammar:

Line 398: is “REF” an abbreviation or does it refer to a missing reference?

Line 438: omit apostrophe on ‘participants’.

Reviewer #2: The authors present an interesting study into how decision making and learning may be affected when it benefits another person rather than oneself, and when being actively observed and evaluated by that other person. They adapted a two-step reinforcement learning task to assess how this social context influences the trade-off between so called “model-free” vs “model-based” learning. Results did not reveal robust effects of social contents on learning. I believe this study could be worth publishing, namely to avoid the file-drawer problem. However, major revisions of the manuscript are needed, as currently there are some theoretical issues that would merit further consideration, and more information in the results would help better evaluate conclusions.

Major issues:

1. Theoretically, I am surprised by the use of the term “vicarious learning” for this context. Most typically, I have found “vicarious learning” to be used in contexts of observational learning, i.e. when learning from observing other’s actions and outcomes, without performing the actions oneself (e.g. Burke et al 2010 PNAS). When one’s actions benefit someone else, that has been called prosocial learning (e.g. Lockwood et al 2016). The current study rather focuses on the latter, where an individual makes choices and observes their outcomes, but sometimes the monetary benefit of the outcomes will revert to another instead of oneself. In other words, a relevant distinction for the processes of reinforcement learning is whether the individual makes the choice/action themselves or only observes the choice/action made by another (which studies have shown engages different neural substrates). Here, the only difference between the conditions is whether I believe that optimising my choices to make the most points will eventually result in more money for me or for another, but in both conditions I get the same information from making choices and observing their outcomes. The information gained and the feedback that I made the right choices is intrinsically rewarding, with the monetary gains being an added bonus that may (or not) motivate me to do even better. I believe these distinctions are worth considering further, for consistency with the terminology in the existing literature, and interpretation of the findings.

2. Related, since the other person was always present, and from the information provided and Figure 1, it seems people were not continuously reminded during the learning task who the outcomes reverted to? If only shown that at the beginning of the block (and given the block design of doing all the “self” vs all the “other” blocks separately), it would implicate further cognitive load to be continuously reminding oneself who the money was for, and the social observation would have already added pressure and load. Thus, people may have quickly forgotten and resumed performing the task as before. (In contrast, e.g. Lockwood et al 2016 showed the beneficiary’s name at the start of every trial and together with the outcome information.)

3. Further details across the article, but particularly in the results section, would help readability and understanding for those in the field, but less familiar with the 2-step task. For example, it can seem confusing to look at the “stay” choice probability, since changes in stimuli pairs mean choosing a different stimulus. It’s unclear what exactly are the measures that constitute the model-free vs model-based “indices”? In page 18, a regression model is described, but the results are not shown (e.g. in a table, or describing the results for all predictors). The paragraph of lines 335-339 may be referring to that model, but that should be made explicit and points accompanied with the statistics. It’s unclear what is the 3rd predictor, of the difference in outcome between chosen and unchosen option? If only showing factual outcomes for the choice made, how is the counterfactual outcome (for the unchosen option) being estimated (namely for the participant)? Statistics are missing for other statements too (e.g. section on BFNE questionnaire), or it unclear what data/tests one is referring to (e.g. lines 352-355, 380-382…).

Minor issues:

1. Data does not seem available on the OSF page provided, which has code for the experiment and analysis, but no data I could find.

2. Typo P.10, line 168: “we decreased from 200ms to 100ms” – that must be 2000ms to 1000ms (2 to 1 secs)?

3. Type P.19, line 346: “was significantly caused participants to choose…” ¬– need to rephrase and not imply causality from a correlation. Likewise, in the next page 20, should rephrase the sentences appropriately for between subjects correlations (i.e. people with higher scores in X show lower scores in y).

4. Page 23, line 414, the figure does not show a “complete crossover” interaction, which implies opposing directions of relation, and the figure just shows two positive relations with different slopes.

6. PLOS authors have the option to publish the peer review history of their article (what does this mean?). If published, this will include your full peer review and any attached files.

Reviewer #1: No

Reviewer #2: No

---

## [Author Response · Author response to Decision Letter 0]

19 May 2023

We appreciate the detailed and constructive comments that we received from you and the reviewers.

We have revised our manuscript regarding the reviewers’ comments. According to Reviewer 2’s suggestion we have changed our terminology from vicarious learning to prosocial learning. We would like to submit our revised manuscript “Prosocial learning: model-based or model-free?” to be considered for publication as a report (research article) in PLoS One.

We therefore would be grateful if you could consider publishing our paper in PLoS One. 

We declare that this manuscript is original, has not been published before, and is not currently being considered for publication elsewhere. We also declare no conflict of interest.

We clarify that the funders (European Research Council, Templeton Trust) had no role at any stage of the current study. 

“Bahador Bahrami is supported by the European Research Council (ERC), under the European Union’s Horizon 2020 research and innovation programme (819040 - acronym: rid-O). Bahador Bahrami is also supported by the Templeton Religion Trust.”

Thank you for your consideration. We look forward to hearing from you.

 

Reviewer #1:

Overall, the manuscript is well-written and clear. Below some feedback on specific sections:

Comment 1: Line 62: it would be helpful if the authors define what a value-based decision is, as this is not clear from the context.

Reply 1: We appreciate the constructive comments of the reviewer. 

We have added this definition for “value-based decision”: In value-based decision-making we choose one option out of many that have different intrinsic values for us. In another world our idiosyncratic values determine which option to choose (Glimcher, 2014). See lines 64-66 and the introduction section.

Comment 2: Line 151: mention inclusion and exclusion criteria specifically.

Reply 2: In the revised methods section (lines 153-161) we have now clarified the inclusion and exclusion criteria.

Comment 3: Overall feedback on Discussion: the discussion is well-written as far as analysing the findings of this study, however what seems to be missing is placing these findings in the context of previous literature. There are also no references to previous work. It would strengthen the paper to discuss the findings in light of similar studies on vicarious learning.

Reply 3: We have now modified the discussion to provide a better coverage of the existing literature. In doing so, we have had to adjust the discussion to accommodate Reviewer 2’s suggestion to change our terminology from vicarious learning to prosocial learning. 

As a result, the term vicarious learning has been removed from the introduction, methods and results. To fulfill reviewer 1’s suggestion, in the revised manuscript Discussion (lines 432-436, 494-527) we define vicarious learning and discuss the previous studies findings in relation to our work. 

Comment 4: Line 410: elaborate on the ‘file drawer’ bias and add a reference.

Reply 4: We appreciate the reviewer for pointing this out. We have elaborated this term and added a reference in the revised discussion section (line 450,451): a tendency to publish the studies that show significant results (Rosenthal, 1979)

Minor comments on grammar:

Comment 5: Line 398: is “REF” an abbreviation or does it refer to a missing reference?

Reply 5: Thanks for pointing this out. Now, they are fixed in the revised manuscript. See lines 435 and the discussion section. 

Comment 6: Line 438: omit apostrophe on ‘participants’.

Reply 6: Thanks for pointing this out. Now, they are fixed in the revised manuscript. See lines 490 and the discussion section. 

 

Reviewer #2:

 The authors present an interesting study into how decision making and learning may be affected when it benefits another person rather than oneself, and when being actively observed and evaluated by that other person. They adapted a two-step reinforcement learning task to assess how this social context influences the trade-off between so called “model-free” vs “model-based” learning. Results did not reveal robust effects of social contents on learning. I believe this study could be worth publishing, namely to avoid the file-drawer problem. However, major revisions of the manuscript are needed, as currently there are some theoretical issues that would merit further consideration, and more information in the results would help better evaluate conclusions.

We appreciate the detailed and constructive comments of the reviewer, acknowledging the value of reporting the negative result as the outcome of study.

Major issues:

Comment 1: Theoretically, I am surprised by the use of the term “vicarious learning” for this context. Most typically, I have found “vicarious learning” to be used in contexts of observational learning, i.e. when learning from observing other’s actions and outcomes, without performing the actions oneself (e.g. Burke et al 2010 PNAS). When one’s actions benefit someone else, that has been called prosocial learning (e.g. Lockwood et al 2016). The current study rather focuses on the latter, where an individual makes choices and observes their outcomes, but sometimes the monetary benefit of the outcomes will revert to another instead of oneself. In other words, a relevant distinction for the processes of reinforcement learning is whether the individual makes the choice/action themselves or only observes the choice/action made by another (which studies have shown engages different neural substrates). Here, the only difference between the conditions is whether I believe that optimising my choices to make the most points will eventually result in more money for me or for another, but in both conditions I get the same information from making choices and observing their outcomes. The information gained and the feedback that I made the right choices is intrinsically rewarding, with the monetary gains being an added bonus that may (or not) motivate me to do even better. I believe these distinctions are worth considering further, for consistency with the terminology in the existing literature, and interpretation of the findings.

Reply 1: We are grateful to the reviewer for this eloquent clarification of the concepts of prosocial and vicarious learning. We have now incorporated this very useful exposition in our paper in the Discussion section (see lines 432-436, 494-527). We have replaced the term “vicarious learning” with “prosocial learning”. Moreover, depending on the relevant content, we have used proxy decision-making, self-other decision-making and prosocial learning interchangeably where the context has allowed it. Please see lines 22, 35, 43, 44, 46, 49, 53, 56, 82-90, 133, 533 in the manuscript. 

Comment 2: Related, since the other person was always present, and from the information provided and Figure 1, it seems people were not continuously reminded during the learning task who the outcomes reverted to? If only shown that at the beginning of the block (and given the block design of doing all the “self” vs all the “other” blocks separately), it would implicate further cognitive load to be continuously reminding oneself who the money was for, and the social observation would have already added pressure and load. Thus, people may have quickly forgotten and resumed performing the task as before. (In contrast, e.g. Lockwood et al 2016 showed the beneficiary’s name at the start of every trial and together with the outcome information.)

Reply 2: We thank the reviewer for the critical evaluation of our experiment structure. Originally, we had reasoned that because we have a block design and there was no consecutive shift between self and other conditions throughout the experiment, therefore individuals would not be expected to experience a further cognitive load in remembering the beneficiary. In addition to reminding the beneficiary to participants before each block, we also reminded them in both conditions that all of the rules were identical between the two conditions except the color of the destination planets. In half of the groups, planets in “self” conditions were yellow and green and in “other” conditions were red and blue and vice versa in the other half. We have now added this explanation to the revised method section (lines 210-215). We have also edited the definition of experiment regarding planets’ color (lines 234-237). Nonetheless, we agree that the issue of cognitive load is a delicate point which might have affected our results. We therefore we have added a statement to our revised discussion section and pointed this issue as a potential factor to be considered in future studies. See lines 461 to 483 and the revised discussion section. Our data is available here: https://osf.io/3k75v/. 

Comment 3: Further details across the article, but particularly in the results section, would help readability and understanding for those in the field, but less familiar with the 2-step task. For example, it can seem confusing to look at the “stay” choice probability, since changes in stimuli pairs mean choosing a different stimulus. It’s unclear what exactly are the measures that constitute the model-free vs model-based “indices”? In page 18, a regression model is described, but the results are not shown (e.g. in a table, or describing the results for all predictors). The paragraph of lines 335-339 may be referring to that model, but that should be made explicit and points accompanied with the statistics. It’s unclear what is the 3rd predictor, of the difference in outcome between chosen and unchosen option? If only showing factual outcomes for the choice made, how is the counterfactual outcome (for the unchosen option) being estimated (namely for the participant)? Statistics are missing for other statements too (e.g. section on BFNE questionnaire), or it unclear what data/tests one is referring to (e.g. lines 352-355, 380-382…).

Reply 3: We appreciate the precise and constructive comments of the reviewer. We have added a paragraph to clarify the definition of “stay” and the measures that constitute the model-free and model-based indices. Please look at lines 245 to 257 and the revised Design section. We have also added statistics for behavioral analysis to further clarification (lines 325-329, 332-336).

The third predictor of the regression model played the same role as the weighting parameter in the computational model and we have explained it more in lines 355 to 359 on the revised Results section. We have added the statistics to this part in terms of the results of the regression model shown in figure 2. Please look at lines 367 to 372 and the revised Results section. We have added a table to show the multilevel logistic regression results. Please see table 1, line 374 on the revised Results section.

We have added statistics for BFNE results. See lines 404 to 407 and the revised Results section for BFNE-II questionnaire. We have specified the data and test of figure 3c which was a repeated factorial measure ANOVA analysis. See lines 388 to 385 and also the caption of figure 3c at lines 396 to 397.

Minor issues:

Comment 4: Data does not seem available on the OSF page provided, which has code for the experiment and analysis, but no data I could find.

2. Typo P.10, line 168: “we decreased from 200ms to 100ms” – that must be 2000ms to 1000ms (2 to 1 secs)?

Reply 4: we have corrected this mistake in the revised methods section (line 174)

Comment 5: Type P.19, line 346: “was significantly caused participants to choose…” ¬– need to rephrase and not imply causality from a correlation. Likewise, in the next page 20, should rephrase the sentences appropriately for between subjects correlations (i.e. people with higher scores in X show lower scores in y).

Reply 5: we have corrected this issues in the revised results section (line 382 and 385)

Comment 6: Page 23, line 414, the figure does not show a “complete crossover” interaction, which implies opposing directions of relation, and the figure just shows two positive relations with different slopes.

Reply 6: Thanks for pointing this out. Now, they have edited this in the revised results section (lines 388-390) and in the revised discussion section (lines 452 to 454).

---

## [Editor Report · Decision Letter 1]

7 Jun 2023

Prosocial learning: model-based or model-free?

PONE-D-22-34158R1

Dear Dr. navidi,

We’re pleased to inform you that your manuscript has been judged scientifically suitable for publication and will be formally accepted for publication once it meets all outstanding technical requirements.

Kind regards,

Francesca Benuzzi, Ph.D.

Academic Editor

PLOS ONE

Additional Editor Comments (optional):

Dear Authors,

I have read with attention your revieers' response and the new version of the manuspcript.

I evalute that with these chsnges the paper is ready for publication in PLOS one.

Sincerly

---

## [Editor Report · Acceptance letter]

14 Jun 2023

PONE-D-22-34158R1 

Prosocial learning: model-based or model-free? 

Dear Dr. Navidi:

I'm pleased to inform you that your manuscript has been deemed suitable for publication in PLOS ONE. Congratulations! Your manuscript is now with our production department. 

Kind regards, 

on behalf of

Professor Francesca Benuzzi 

Academic Editor

PLOS ONE